# Reevaluation of Piezo1 as a gut RNA sensor

Alec R Nickolls[1], Gabrielle S O'Brien[1], Sarah Shnayder[1], Yunxiao Zhang[2], Maximilian Nagel[1], Ardem Patapoutian[2], Alexander T Chesler[1]*

[1]National Center for Complementary and Integrative Health, National Institutes of Health, Bethesda, United States; [2]Department of Neuroscience, Howard Hughes Medical Institute, Dorris Neuroscience Center, Scripps Research Institute, San Diego, United States

**Abstract** Piezo1 is a stretch-gated ion channel required for mechanosensation in many organ systems. Recent findings point to a new role for Piezo1 in the gut, suggesting that it is a sensor of microbial single-stranded RNA (ssRNA) rather than mechanical force. If true, this would redefine the scope of Piezo biology. Here, we sought to replicate the central finding that fecal ssRNA is a natural agonist of Piezo1. While we observe that fecal extracts and ssRNA can stimulate calcium influx in certain cell lines, this response is independent of Piezo1. Additionally, sterilized dietary extracts devoid of gut biome RNA show similar cell line-specific stimulatory activity to fecal extracts. Together, our data highlight potential confounds inherent to gut-derived extracts, exclude Piezo1 as a receptor for ssRNA in the gut, and support a dedicated role for Piezo channels in mechanosensing.

## Editor's evaluation

This is an important study that resolves a controversy about a proposed molecular linkage between the fields of mechanobiology and RNA signaling. While prior research had claimed that a specific mechanosensitive ion channel in the gut responds to a specific fecal RNA, this study provides compelling evidence that the mechanosensitive ion channel does not respond to the RNA.

*For correspondence:
alexander.chesler@nih.gov

## Introduction

Piezo proteins are mechanically gated ion channels that transduce changes in plasma membrane tension into electrical current (*Ridone et al., 2019*; *Szczot et al., 2021*). There are two members in the mammalian Piezo family: Piezo1 and Piezo2 (*Coste et al., 2010*). Piezo1 is expressed in many tissues, including the cardiovascular, hematopoietic, and skeletal systems (*Jiang et al., 2021*; *Li et al., 2014*; *Retailleau et al., 2015*; *Rode et al., 2017*; *Sun et al., 2019*; *Wang et al., 2016*). Specifically, Piezo1 mediates the mechanical sensing of fluid flow and is required for normal development and function of lymph and blood vessels (*Li et al., 2014*; *Liu et al., 2020*; *Nonomura et al., 2018*; *Ranade et al., 2014b*). Additionally, many blood cell types depend on Piezo1 for shear force sensing and volume regulation (*Cahalan et al., 2015*; *Cinar et al., 2015*; *Faucherre et al., 2014*; *Solis et al., 2019*). These vascular functions are particularly evident in the clinical manifestation of Piezo1 mutations in humans (*Albuisson et al., 2013*; *Bae et al., 2013*; *Fotiou et al., 2015*; *Glogowska et al., 2017*; *Lukacs et al., 2015*). Piezo1 also regulates the formation and maintenance of bone and cartilage through mechanical load sensing (*Hendrickx et al., 2021*; *Lee et al., 2014*; *Li et al., 2019*; *Sun et al., 2019*; *Wang et al., 2020*; *Zhou et al., 2020*). In other organ systems, the role of Piezo1 is currently less well-defined, but it is known to function as a mechanosensor in neural stem cells and certain epithelial and mesenchymal cell types (*Eisenhoffer et al., 2012*; *Martins et al., 2016*; *Miyamoto et al., 2014*;

*Pathak et al., 2014*; *Sugimoto et al., 2017*). By contrast, the homologous Piezo2 is expressed at its highest levels in the somatosensory system, vagal-nodose complex, and specialized epithelial cells (*Chiu et al., 2014*; *Kupari et al., 2019*; *Nguyen et al., 2017*; *Usoskin et al., 2015*; *Wang et al., 2017*; *Woo et al., 2014*). Piezo2 is responsible for detecting gentle touch, vibration, and proprioception in mice and humans (*Chesler et al., 2016*; *Ranade et al., 2014a*; *Woo et al., 2015*).

Many extrinsic factors can influence Piezo mechanosensitivity, including membrane tension, membrane voltage, cytoskeletal integrity, extracellular matrix contact, cyclic adenosine monophosphate signaling, and phosphatidylinositol second messenger pathways (*Borbiro et al., 2015*; *Dubin et al., 2012*; *Gaub and Müller, 2017*; *Moroni et al., 2018*; *Narayanan et al., 2018*; *Romero et al., 2019*; *Romero et al., 2020*). In addition to this list, it stands to reason that there could be ligands of Piezo channels that modulate or even directly evoke channel gating. Indeed, high-throughput drug screens have generated the synthetic small molecules called Yoda1, Jedi1, and Jedi2, which act as allosteric modulators of Piezo1 by stabilizing its open conformation (*Botello-Smith et al., 2019*; *Syeda et al., 2015*; *Wang et al., 2018*).

A recent study provided the first evidence of endogenous ligands for Piezo channels (*Sugisawa et al., 2020*). Remarkably, it was shown that Piezo1 functions as a ligand-gated ion channel to sense single-stranded RNAs (ssRNAs). It was proposed that bioactive ssRNAs are produced by the gut microbiome, and that these molecules function through Piezo1 channels to alter serotonin production and trigger a physiological cascade that impacts bone homeostasis. Considering that ssRNAs might be generated and released under a variety of circumstances, the scientific field will have to radically rethink the role of Piezo1 not only in the gut but throughout the body. It also suggests that perhaps Piezo2 can be gated by these types of molecules. Piezo2 is expressed by sensory and vagal neurons targeting the skin and other organ systems with diverse microbiomes that are also sites of viral infections and colonization by pathogenic bacteria or fungi (*Chiu et al., 2014*; *Kupari et al., 2019*; *Nguyen et al., 2017*; *Usoskin et al., 2015*; *Wang et al., 2017*; *Woo et al., 2014*). Therefore, we set out to use calcium imaging and electrophysiological recordings to investigate how ssRNAs derived from the gut influence Piezo channel function.

## Results

### ssRNA40 does not alter calcium activity or mechanotransduction in N2a cells

ssRNA40 is a synthetic 20-mer ssRNA oligonucleotide derived from the human immunodeficiency virus (HIV) genome (*Heil et al., 2004*). ssRNA40 is classically known as an agonist for the immune surveillance toll-like receptors 7 and 8 in mice and humans, respectively (*Heil et al., 2004*; *Zhang et al., 2018*). However, it was recently reported that ssRNA40 also shows agonist activity toward the mechanosensitive ion channel Piezo1 (*Sugisawa et al., 2020*). To test this finding, we first used the Neuro-2a (N2a) mouse neuroblastoma cell line since these cells natively express Piezo1 (*Coste et al., 2010*) and were reported to conduct a Piezo1-dependent current in the presence of ssRNA40 (*Sugisawa et al., 2020*).

For measuring Piezo1 activity during exposure to ssRNA40, we performed in vitro fluorescent calcium imaging on N2a cell cultures with the Fluo-4 AM ester dye. Using ssRNA40 from the same supplier and dosage as prior studies (10 µg/mL) (*Sugisawa et al., 2020*), we found that application of ssRNA40 did not elicit a detectable increase in fluorescence, even when the imaging time frame was extended up to 3 min (*Figure 1A* and *Video 1*). As a positive control for Piezo1 activation, we used Yoda1, which is known to induce Piezo1-dependent calcium transients (*Syeda et al., 2015*). A 30 µM Yoda1 triggered a dramatic calcium influx over the course of ~1 min in virtually all N2a cells (*Figure 1A–C*). This was followed by application of ionomycin, a potent calcium ionophore, as a further positive control to elicit maximal calcium influx in all cells. For negative controls, we tested the vehicle solution alone to observe mechanosensitive flow responses and also tested ssRNA41 to examine sequence-specific effects of ssRNA (*Figure 1A–C*). ssRNA41 is the same length and sequence as ssRNA40, except all uridine residues are substituted with adenosine; unlike ssRNA40, this molecule is not a TLR7/8 agonist (*Heil et al., 2004*). After autofluorescence subtraction, there was no significant difference in N2a cell calcium activity between these negative controls and ssRNA40 (*Figure 1C*).

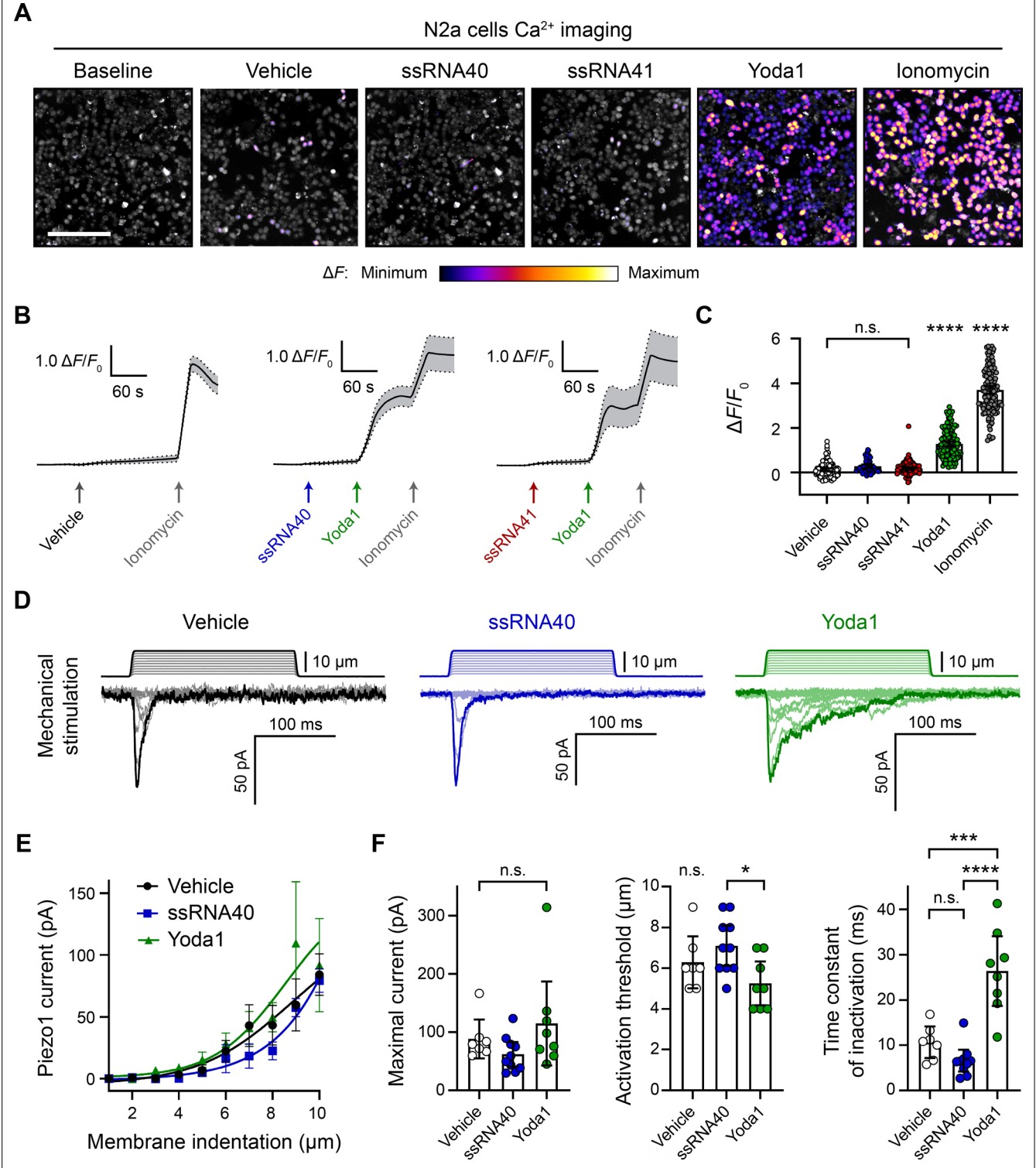

**Figure 1.** ssRNA40 does not alter calcium activity or mechanotransduction in N2a cells. (**A**) Fluo-4 calcium imaging of N2a cells during exposure to different treatments, representative of ≥3 independent recordings for each condition. The magnitude of the change in fluorescence ($\Delta F$) is represented on a fire color scale and is superimposed on a grayscale baseline fluorescence image. Cells were exposed to buffer only (vehicle) or 10 µg/mL ssRNA40 or ssRNA41 for up to 3 min, followed by 30 µM Yoda1 and 10 µM ionomycin. Scale bar is 200 µm. (**B**) Example calcium imaging traces of vehicle, ssRNA40, or ssRNA41, as well as Yoda1 and ionomycin control treatments. Fluorescence values are shown as $\Delta F$ normalized to the initial fluorescence ($\Delta F/F_0$). n = 50 cells plotted as mean ± 95% CI. (**C**) Quantification of calcium responses to different treatments. n = 50 cells per condition. Error bars indicate mean ± 95% CI. One-way ANOVA with Bonferroni correction: not significant (n.s.) p≥0.05, **** p<0.0001. (**D**) Example whole-cell voltage-

*Figure 1 continued on next page*

*Figure 1 continued*

clamp recordings of N2a cells during mechanical stimulation. Top traces indicate the magnitude of plasma membrane indentation in 1 µm steps, and bottom traces show whole-cell currents elicited by the stimuli. Vehicle, 10 µg/mL ssRNA40, or 30 µM Yoda1 were bath-applied 10 min prior to recording. (**E**) Piezo1 current versus membrane indentation to illustrate stimulus–response relationships. n = 7–10 cells per condition are plotted to show the mean current per level of indentation, with error bars indicating the standard error of the mean (SEM). (**F**) Quantification of mechanically evoked current amplitude, threshold, and inactivation. n = 7–10 cells per condition. Error bars represent mean ± 95% CI. One-way ANOVA with Bonferroni correction: n.s. p≥0.05, *p<0.05, ***p<0.001, ****p<0.0001.

In addition to directly activating Piezo1, ssRNA40 was reported to delay the inactivation of Piezo1 mechanically evoked currents (*Sugisawa et al., 2020*), similar to Yoda1 (*Syeda et al., 2015*). We tested this claim by whole-cell voltage-clamp recordings of N2a cells during simultaneous mechanical stimulation of the plasma membrane. Mechanical stimuli were administered with a nanomotor probe to indent the cell surface in 1 µm increments (*Video 2*). The stimulation elicited a rapidly inactivating inward current, which is characteristic of the Piezo family of ion channels (*Figure 1D*). As expected, including 30 µM Yoda1 in the external bath solution resulted in a reduced apparent mechanical activation threshold and a prolonged inactivation phase of the currents (*Figure 1D–F*). By contrast, 10 µg/mL ssRNA40 in the bath solution showed no measurable change from the vehicle control on the amplitude, activation threshold, or inactivation rate of mechanically evoked currents (*Figure 1D–F*).

## ssRNA40 does not activate *Piezo1*-transfected HEK293 cells

Compared to the Piezo1 agonist activity of Yoda1, ssRNA40 was reported to have a relatively small effect size (*Sugisawa et al., 2020*). We were concerned that such a small effect size could have been overlooked in our initial experiments, given the relatively low functional expression of Piezo1 in N2a cells. Therefore, we also examined the effects of ssRNA on Piezo1 channels expressed at high levels via transient transfection of human embryonic kidney 293 (HEK293) cells. We performed calcium imaging to compare the responses of native versus *Piezo1*-transfected HEK293 cells during exposure to vehicle, ssRNA40, or ssRNA41 (*Figure 2A and B*). Each imaging trial was followed by stimulation with Yoda1 and ionomycin as positive controls for Piezo1 response and maximal response, respectively. There was a noticeable but nonsignificant calcium response of untransfected cells to Yoda1 (*Figure 2C*), consistent with a previous report that HEK293 cells express very low but detectable levels of human

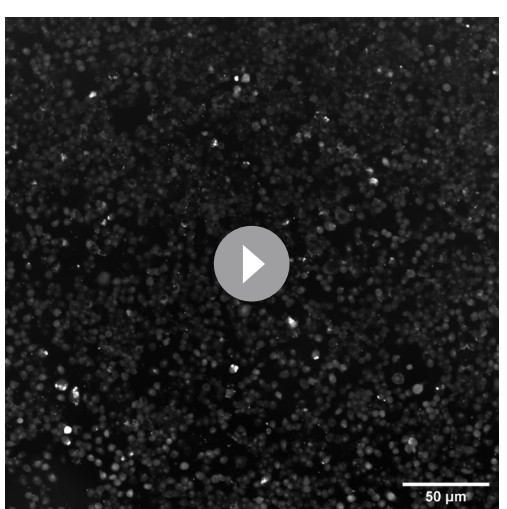

**Video 1.** N2a cells do not respond to ssRNA40. 5 min time-lapse recording of fluo-4 AM fluorescence in N2a cells during sequential exposure to 10 µg/mL ssRNA40, 30 µM Yoda1, and 10 µM ionomycin. 1 s of video is equivalent to 30 s of real time.

https://elifesciences.org/articles/83346/figures#video1

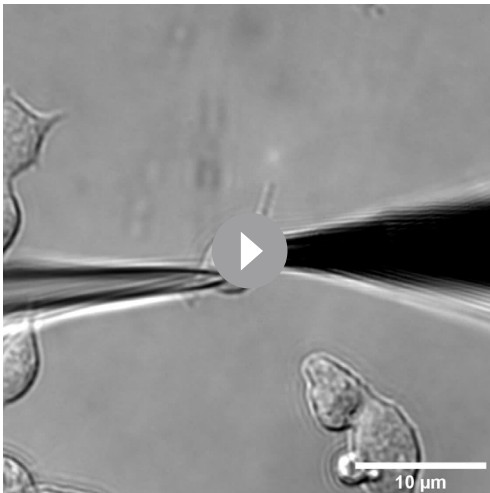

**Video 2.** Mechanical stimulation assay. A HEK293 cell during simultaneous mechanical stimulation and whole-cell current recording. The patch pipette (left) is sealed onto the plasma membrane, and the mechanical probe (right) indents the cell membrane to evoke Piezo1 activity. The video depicts a single 5 µm indentation as part of a larger train of step-wise indentations from 1 to 10 µm.

https://elifesciences.org/articles/83346/figures#video2

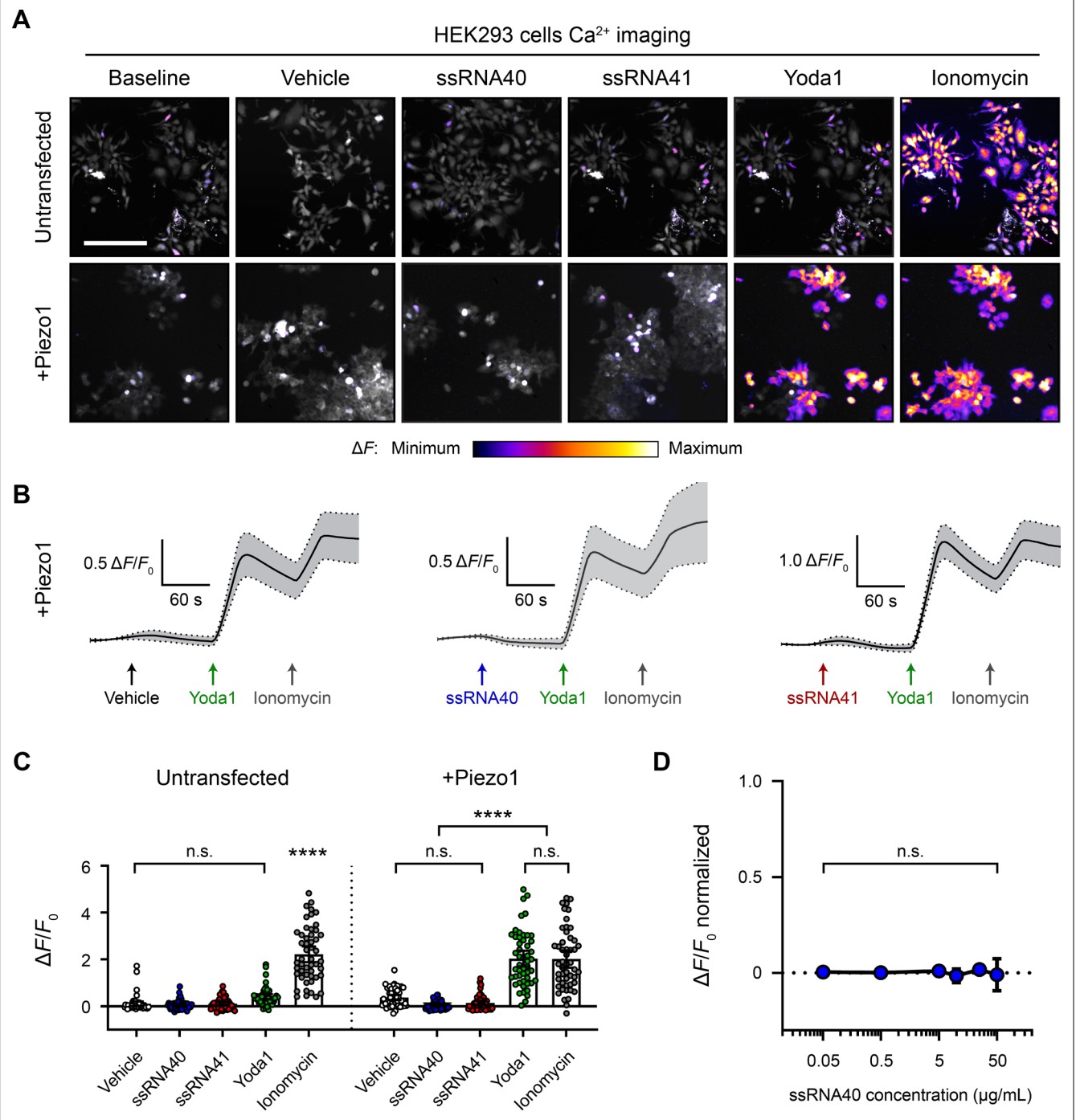

**Figure 2.** ssRNA40 does not activate *Piezo1*-transfected HEK293 cells. (**A**) Fluo-4 calcium imaging of HEK293 cells, with or without transfection of mouse *Piezo1*, representative of ≥3 independent recordings for each condition. Treatment concentrations are 10 µg/mL ssRNA40 or ssRNA41, 30 µM Yoda1, and 10 µM ionomycin. Scale bar is 200 µm. (**B**) Example calcium imaging traces of *Piezo1*-transfected HEK293 cells during different treatments. Yoda1 was applied 90 s after any given RNA sample, and only cells that responded to Yoda1 (presumably Piezo1-transfected) were analyzed. Transfection efficiency was generally >60% of the cell culture. n = 50 cells plotted as mean ± 95% CI. (**C**) Quantification of HEK293 cell calcium responses. n = 50 cells per condition plotted as mean ± 95% CI. One-way ANOVA with Bonferroni correction: n.s. p≥0.05, ****p<0.0001. (**D**) Dose–response curve of ssRNA40 treatment on *Piezo1*-transfected GCaMP6s-expressing HEK293 cells. After 1 min of baseline measurement, ssRNA40 was administered for 3 min followed by ionomycin for 1 min. A random selection of cells was analyzed from each recording. The responses are normalized, with the ionomycin response being $\Delta F/F_0 = 1$. n = 25 cells per dose plotted as mean ± 95% CI. One-way ANOVA with Bonferroni correction: n.s. p≥0.05.

The online version of this article includes the following figure supplement(s) for figure 2:

**Figure supplement 1.** ssRNA40 does not activate Piezo1 or modify its response to Yoda1.

Piezo1 (*Dubin et al., 2017*). However, in both untransfected and *Piezo1*-transfected conditions, we found no significant calcium activity between vehicle, ssRNA40, and ssRNA41 (*Figure 2C*).

Prior studies used between 5 and 20 µg/mL ssRNA40 in their experiments (*Heil et al., 2004*; *Lehmann et al., 2012*; *Shibata et al., 2016*; *Sugisawa et al., 2020*). For that reason, our above experiments used 10 µg/mL ssRNA40. However, to explore whether ssRNA40 has a dose–response effect on Piezo1, we performed calcium imaging on *Piezo1*-transfected HEK293 cells during exposure to different concentrations of ssRNA40. Using a log scale range from 0.05 to 50 µg/mL, we did not observe significant calcium influx triggered by ssRNA40 at any concentration (*Figure 2D*).

An inability to replicate a finding could reflect unappreciated nuances in how two groups conduct the experiments. To address this possibility, we tested whether ssRNA40 could activate Piezo1 in a completely independent laboratory using a distinct cell line, a different methodology, and separately sourced reagents. For these studies, we used a Piezo1-knockout (Piezo1-KO) genetic background HEK293 cell line (*Dubin et al., 2017*). Importantly, this is the identical cell line used in the originally published ssRNA experiments on Piezo1 (*Sugisawa et al., 2020*). We recorded the calcium response to ssRNA40 with or without *Piezo1* transfection using a fluorescence imaging plate reader (FLIPR) calcium flux screening platform. Several ssRNA40 concentrations were tested from 0.312 to 5 µg/mL followed by Yoda1 as a positive control. We compared untransfected cells to cells transfected with either mouse *Piezo1* or human *Piezo1* (*Figure 2—figure supplement 1*). No increase in calcium was detected throughout exposure to ssRNA40, and there were no significant differences between transfected and untransfected cells or across the various ssRNA40 concentrations (*Figure 2—figure supplement 1*). To examine whether ssRNA40 might more subtly potentiate Piezo1 activity, we checked whether the response to 5 µM Yoda1 was increased after exposure to ssRNA40. However, we found no change in the Yoda1 response following treatment with vehicle or different ssRNA40 concentrations (*Figure 2—figure supplement 1*). Together, these data independently corroborate our previous observations that ssRNA40 does not activate Piezo1 directly or modulate its mechano-transduction in either N2a cells or HEK293 cells.

## Fecal and dietary extracts activate HEK293 cells independently of Piezo1

The concept that ssRNA can activate Piezo1 originally arose from screening components of mouse feces and led to the hypothesis that compounds produced by the gut microbiome directly influence Piezo1 function. It was reported that both crude fecal extracts and purified fecal RNA elicit a calcium influx in *Piezo1*-transfected HEK293 cells (*Sugisawa et al., 2020*). Although the synthetic ssRNA40 molecule did not activate Piezo1 in our hands, it remained possible that fecal preparations could show agonist activity. We homogenized and diluted mouse fecal matter to 100 mg/mL and filtered it through a 0.45 µm mesh to eliminate any undissolved sample. Pipetting this solution onto cells resulted in extremely high autofluorescence that precluded calcium imaging. However, we found that the autofluorescence was mostly eliminated if the feces were diluted to at least 5 mg/mL – we refer to this diluted filtered sample as 'fecal extract.' Applying fecal extract to *Piezo1*-transfected HEK293 cells triggered a substantial calcium response (*Figure 3A* and *Video 3*). More dilute preparations of extract yielded little or no calcium influx. Interestingly, across several imaging trials, we noticed that the response to 5 mg/mL fecal extract was variable; not all cells in the field of view would necessarily respond, and the response onset was often 30–60 s after the fecal extract was first added (*Figure 3A and B*). Notably, however, we also found that untransfected HEK293 cells had a similar calcium response to fecal extract, suggesting that fecal extracts may trigger calcium influx via a Piezo1-independent mechanism (*Figure 3A and B*). This was confirmed by applying fecal extract to Piezo1-KO HEK293 cells, which still responded despite the complete absence of Piezo1 (*Figure 3—figure supplement 1*).

Crude fecal extracts are complex biochemical mixtures that include products from the microbiome. To test the contribution of bacterial RNA in the calcium response of HEK293 cells to fecal extract, we purified these nucleic acids (*Figure 3C*) from our samples and tested whether they activated Piezo1. In contrast to the total extract, we did not observe a detectable calcium influx with 10 µg/mL fecal RNA – the same concentration used in other studies (*Figure 3D*; *Sugisawa et al., 2020*). We additionally extracted fecal RNA using the same kit and procedure as prior studies (*Sugisawa et al., 2020*), but this sample likewise failed to induce any activity (see 'Materials and methods'). We then performed the reciprocal experiment by treating fecal extracts with the ssRNA-degrading enzyme

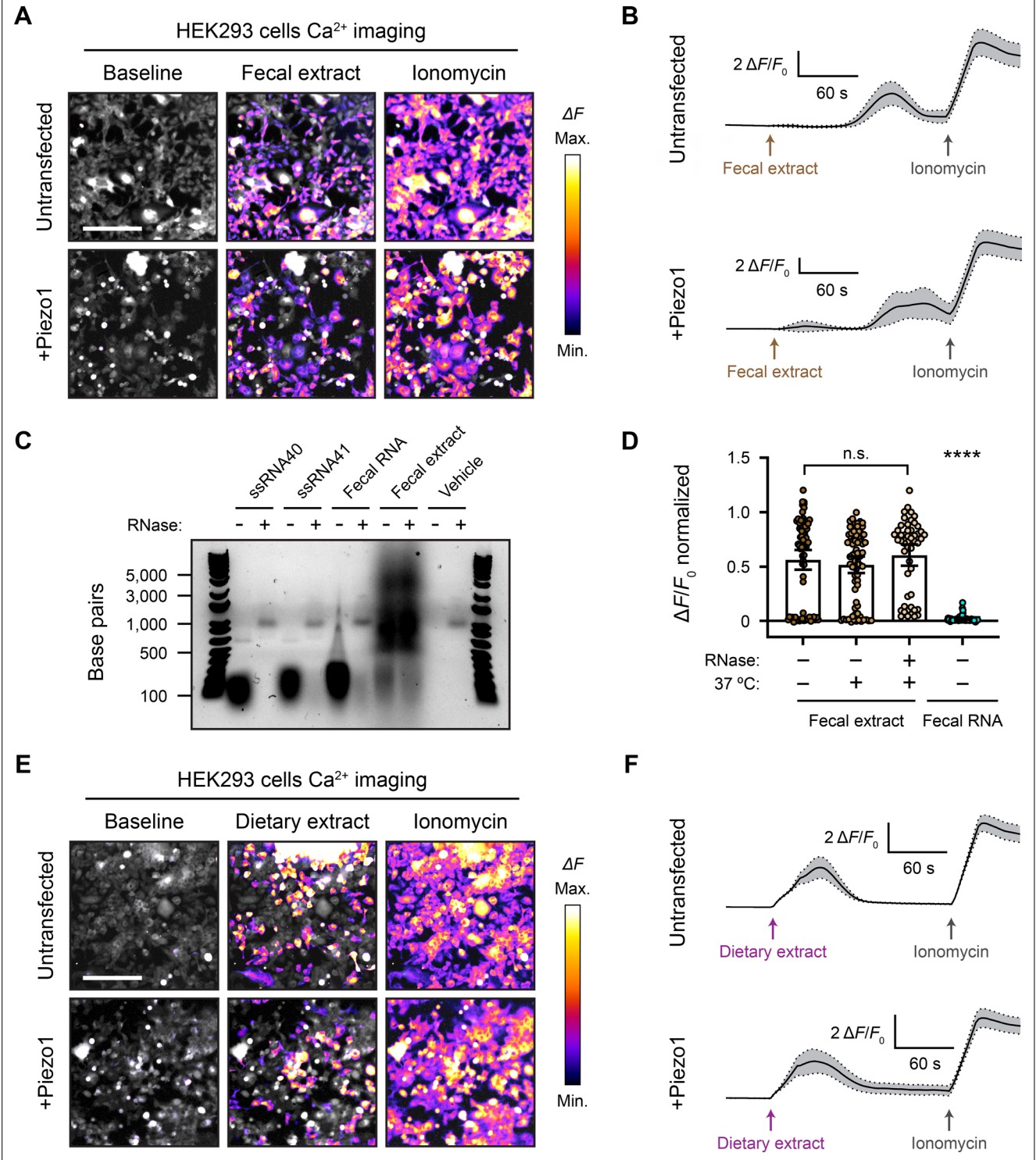

**Figure 3.** Fecal and dietary extracts activate HEK293 cells independently of Piezo1. (**A**) GCaMP6s calcium imaging of HEK293 cells during exposure to 5 mg/mL fecal extract, with or without *Piezo1* transfection, representative of ≥3 independent recordings for each condition. Scale bar is 200 μm. (**B**) Example calcium imaging traces of HEK293 cell responses to fecal extract. n = 50 cells per condition plotted as mean ± 95% CI. (**C**) Agarose gel showing the nucleic acid content of 50 μg/mL purified fecal RNA and 100 mg/mL crude fecal extract. 50 μg/mL of ssRNA40 and ssRNA41 were used as positive controls since they are pure RNA samples of a defined mass and sequence. Ringer's solution was used as vehicle negative control. Treating the samples with RNase A eliminated the low-molecular-weight nucleic acid (<500 bp). The crude fecal extract additionally had a high-molecular-weight smear (500–5000 bp) that was unaffected by RNase A treatment, which is likely DNA. (**D**) Quantification of *Piezo1*-transfected HEK293 cell responses to 5 mg/mL fecal extracts that were untreated (control), heat-treated (mock), or heat + RNase A-treated (RNase), as well as 10 μg/mL fecal RNA. The $\Delta F/F_0$

*Figure 3 continued on next page*

*Figure 3 continued*

values are normalized, with the ionomycin response being $\Delta F/F_0 = 1$. n = 50 cells per condition plotted as mean ± 95% CI. Kruskal–Wallis with Dunn's multiple-comparisons test: n.s. p≥0.05, ****p<0.0001. (**E**) GCaMP6s calcium imaging of HEK293 cells during exposure to 5 mg/mL dietary extract, with or without *Piezo1* transfection, representative of ≥2 independent recordings for each condition Scale bar is 200 µm. (**F**) Example calcium imaging traces of HEK293 cell responses to dietary extract. n = 50 cells per condition plotted as mean ± 95% CI.

The online version of this article includes the following source data and figure supplement(s) for figure 3:

**Source data 1.** Original uncropped RNA gel RNA and fecal samples, untreated or RNase-treated, separated on a 1% agarose gel to examine the RNA content and effect of RNase on the samples.

**Figure supplement 1.** Fecal and dietary extracts activate HEK293 Piezo1-KO cells.

RNase A. Both RNase-treated and untreated fecal extracts activated *Piezo1*-transfected HEK293 cells to a similar degree (***Figure 3D***). Together, these experiments demonstrate that fecal extracts can stimulate calcium influx in HEK293 cells, but this effect is unlikely to be mediated by RNA and, regardless, the activity does not depend on Piezo1.

To disentangle possible sources of activation in our fecal extracts, we evaluated whether anything in the mouse diet might stimulate HEK293 cell calcium influx. We reasoned that separately testing the food input would eliminate host and microbial factors found in the fecal output. Mouse chow pellets were dissolved and filtered to prepare a dietary extract by the same method as the previous fecal extracts. Surprisingly, applying 5 mg/mL dietary extract to HEK293 cells produced a substantial calcium response (***Figure 3E***). This activation occurred in both untransfected and *Piezo1*-transfected cells. Similar to fecal extract, the dietary extract often activated only a subset of cells (***Figure 3E*** and ***Video 4***). The timing of the response onset was also variable but tended to occur earlier compared to fecal extract (***Figure 3F***). Dietary extract also activated Piezo1-KO HEK293 cells, confirming that Piezo1 is dispensable for the response to the extract (***Figure 3—figure supplement 1***). To verify that the dietary and fecal extracts were not activating cells in a nonspecific way by changing the osmolality or pH, we confirmed that these properties were not substantially altered between control solutions and the crude extracts (see 'Materials and methods'). These results suggest that an element of the mouse diet, if present in fecal matter, could be a confounding factor when studying active compounds derived from the host gut or resident microbiota in calcium imaging assays.

Piezo1 is a nonselective ion channel that inactivates quickly (within 10–30 ms) (***Coste et al., 2010***). Therefore, calcium imaging is not the most sensitive readout of channel gating. To more definitively confirm that fecal extracts and ssRNA are not affecting Piezo1 activity, we carried out a series of

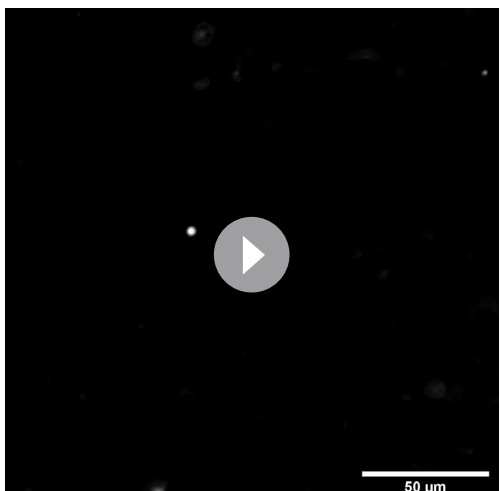

**Video 3.** HEK293 cells respond to fecal extract. 5 min time lapse of GCaMP6s fluorescence in HEK293 cells during exposure to 5 mg/mL fecal extract and then 10 µM ionomycin. 1 s of video is equivalent to 30 s of real time.

https://elifesciences.org/articles/83346/figures#video3

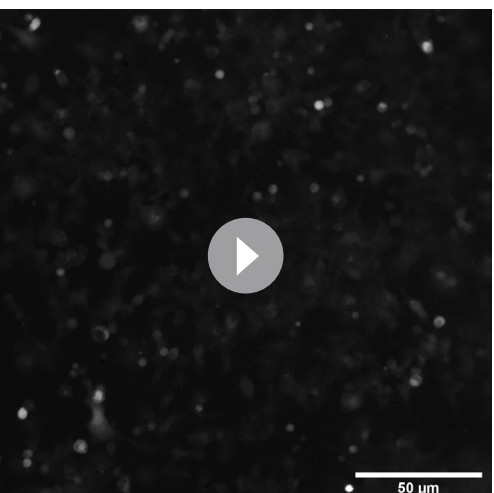

**Video 4.** HEK293 cells respond to dietary extract. 5 min time lapse of GCaMP6s fluorescence in HEK293 cells during exposure to 5 mg/mL dietary extract and then 10 µM ionomycin. 1 s of video is equivalent to 30 s of real time.

https://elifesciences.org/articles/83346/figures#video4

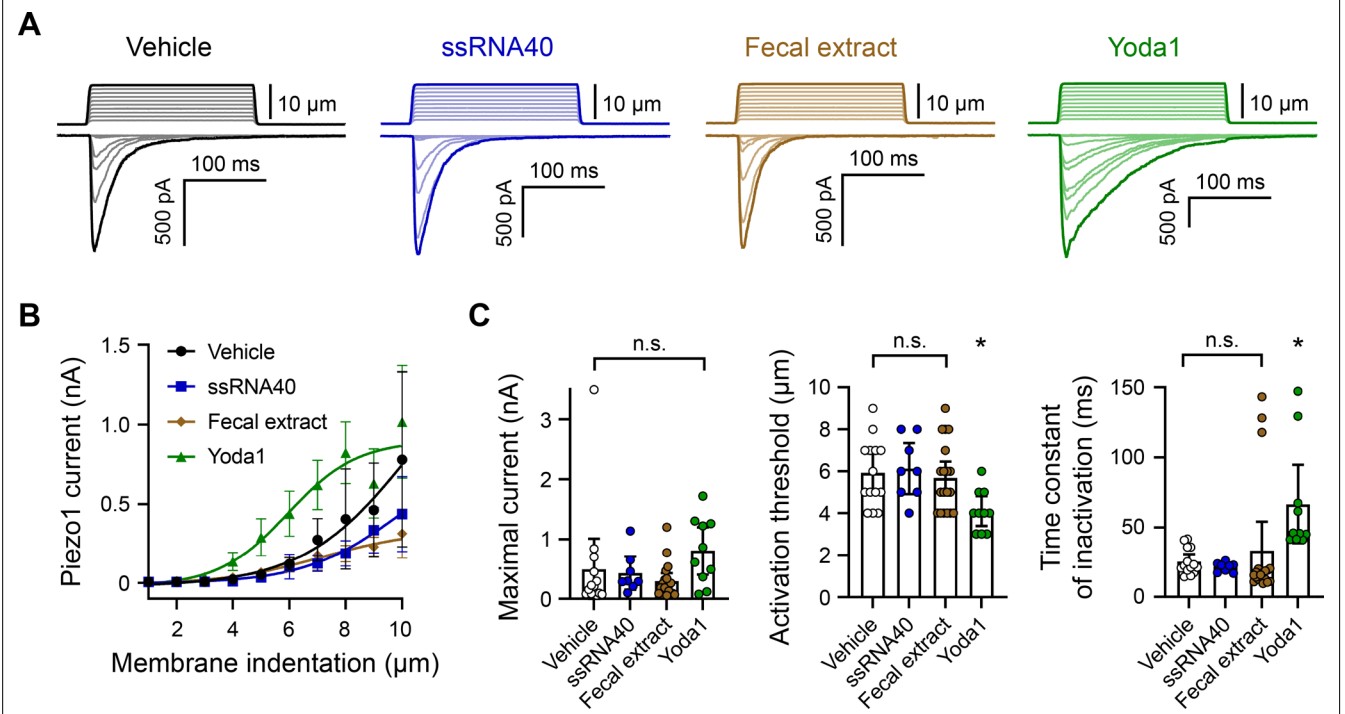

**Figure 4.** ssRNA40 and fecal extract do not modify Piezo1 mechanotransduction. (**A**) Example whole-cell voltage-clamp recordings of *Piezo1*-transfected HEK293 cells during mechanical stimulation. Top traces indicate the magnitude of plasma membrane indentation in 1 µm steps, and bottom traces show whole-cell currents elicited by the stimuli. Vehicle, 10 µg/mL ssRNA40, 5 mg/mL fecal extract, or 30 µM Yoda1 were bath-applied 5 min prior to recording. (**B**) Piezo1 current versus membrane indentation demonstrates the stimulus–response relationships across conditions. n = 8–19 cells per condition are shown as mean ± SEM. (**C**) Quantification of mechanically evoked current amplitude, threshold, and inactivation. Individual cell responses are plotted: n = 14 vehicle, n = 8 ssRNA40, n = 19 fecal extract, and n = 10 Yoda1. Error bars represent mean ± 95% CI. One-way ANOVA with Bonferroni correction: n.s. p≥0.05, *p<0.05.

whole-cell voltage-clamp recordings on *Piezo1*-transfected HEK293 cells. The cells were mechanically stimulated in the presence of ssRNA40 or fecal extract (*Figure 4A*). As negative and positive controls for Piezo1 agonism, vehicle solution and Yoda1 were used, respectively. Across all conditions, the maximum whole-cell Piezo1 current evoked by mechanical stimulation was not substantially different. By contrast, Yoda1 significantly lowered the apparent mechanical threshold for Piezo1 activation and delayed channel inactivation as expected (*Figure 4B and C*; *Syeda et al., 2015*). Together, our results do not provide evidence for even a limited effect of either ssRNA40 or fecal extracts on the biophysical properties of Piezo1.

## Calcium response to fecal and dietary extracts is cell line-specific

We were concerned about the potential confounding effects of fecal/dietary extract-induced activation of HEK293 cells and wondered whether this effect extended to other commonly used cell lines in the field. Piezo1 was originally discovered in N2a cells (*Coste et al., 2010*), and this cell line continues to be frequently used for in vitro work on Piezo1 (*Geng et al., 2020*; *Ridone et al., 2020*; *Romero et al., 2019*). Therefore, we compared N2a and HEK293 cells, with or without *Piezo1* transfection, during treatment with fecal/dietary extracts. Since Piezo1 is endogenously expressed in wildtype N2a cells, we performed these experiments on a Piezo1-KO N2a cell line (*Moroni et al., 2018*).

The untransfected N2a Piezo1-KO cells showed no calcium response to Yoda1 treatment, confirming an absence of Piezo1 (*Figure 5A*). Conversely, N2a Piezo1-KO cells that were transfected with *Piezo1* showed a large calcium response to Yoda1, confirming an efficient transfection. Interestingly, unlike HEK293 cells, N2a cells did not respond to either fecal or dietary extracts (*Figure 5A*). Moreover, *Piezo1* transfection did not endow N2a cells with sensitivity to either of these extracts (*Figure 5B*). From these data, we conclude that neither fecal extracts nor RNA derived from the gut microbiome

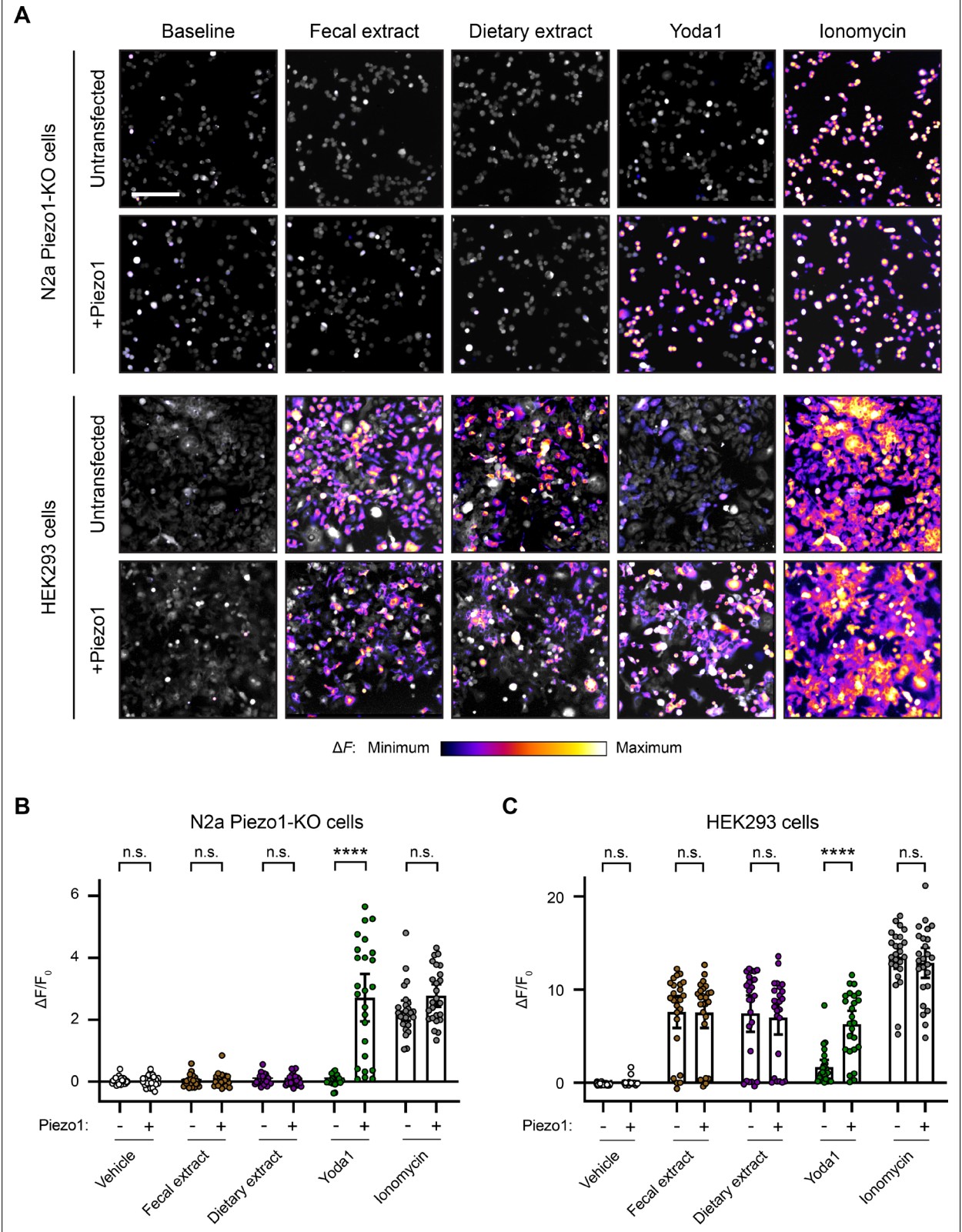

**Figure 5.** Fecal and dietary extracts induce cell line-specific activity independently of Piezo1. (**A**) Calcium imaging of N2a Piezo1-KO cells and HEK293 cells, with or without *Piezo1* transfection, representative of ≥2 independent recordings for each condition. Fluo-4 or GCaMP6s were used to image the N2a cells or HEK293 cells, respectively. Treatment concentrations are 5 mg/mL fecal or dietary extract, 30 μM Yoda1, or 10 μM ionomycin. Scale bar is 200 μm. (**B**) Quantification of calcium responses. n = 25 cells per condition plotted as mean ± 95% CI. Pairwise comparisons between untransfected and transfected recordings using Kruskal–Wallis with Dunn's multiple-comparisons test: n.s. p≥0.05, **** p<0.0001.

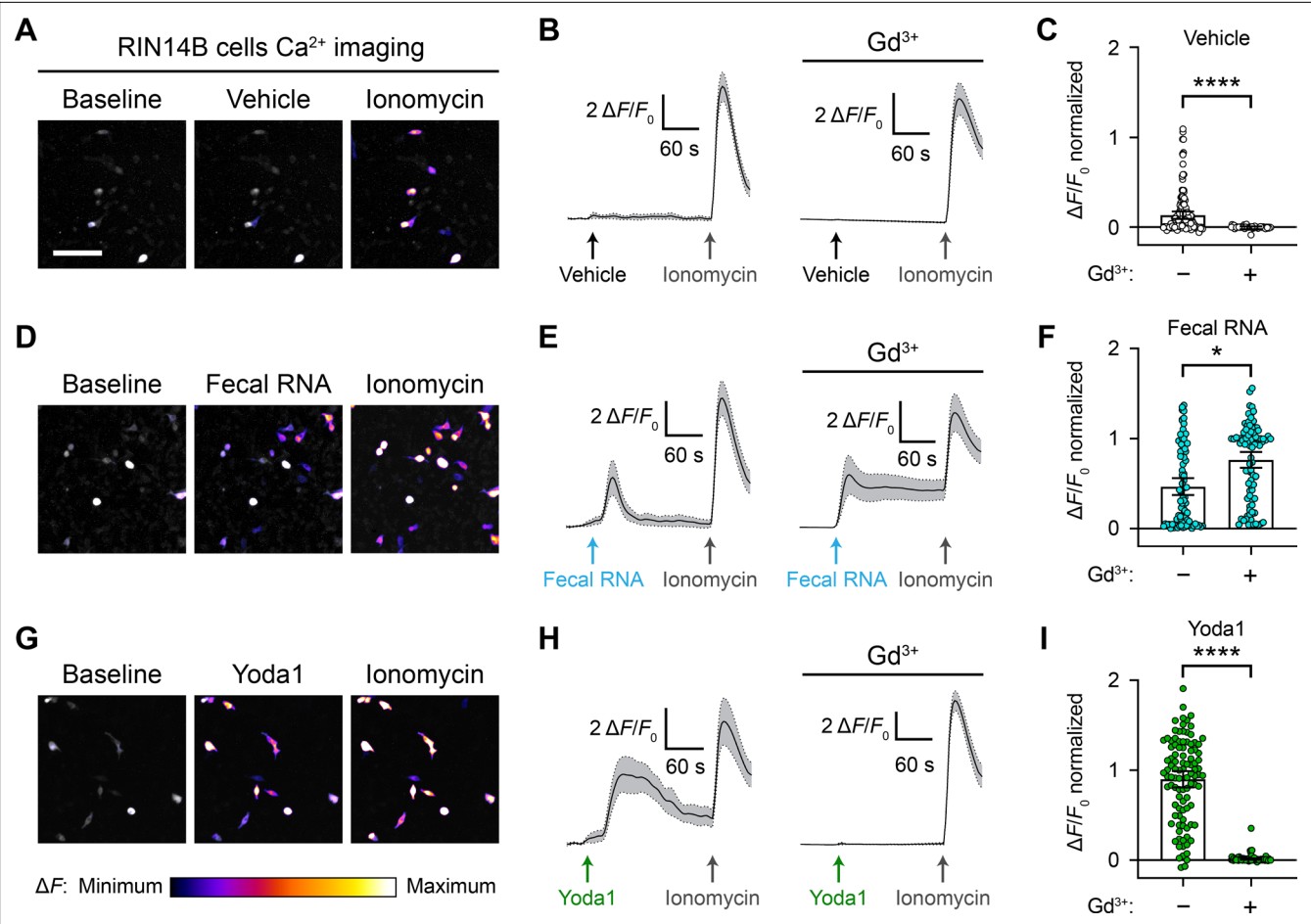

**Figure 6.** RNA activates RIN14B cells independently of Piezo1. (**A–C**) Calcium imaging of RIN14B cell activity during application of negative control vehicle with and without gadolinium inhibition of Piezo1. Gadolinium visibly reduced spontaneous calcium transients. (**D–F**) RIN14B cell calcium influx in response to fecal RNA, with and without gadolinium. (**G–I**) RIN14B cell calcium influx in response to the positive control Piezo1 agonist Yoda1, which is blocked by gadolinium. The calcium imaging was performed on GCaMP6s-transfected cells. GCaMP6s calcium responses were measured during stimulation with 25 μg/mL fecal RNA, 15 μM Yoda1, and 10 μM ionomycin. To block Piezo1, 30 μM gadolinium was preincubated on the cells for 5 min and included throughout the calcium imaging recording. Line graphs represent mean ± 95% CI of a single recording each of n = 50 cells. Bar graphs represent n = 100–150 cells from ≥2 independent recordings for each condition, with fluorescence values normalized to the response to ionomycin = 1.0, and the bars indicate mean ± 95% CI. Pairwise comparisons between untreated and gadolinium (Gd3+)-treated recordings using Kruskal–Wallis with Dunn's multiple-comparisons test: *p<0.05, ****p<0.0001. The scale bar for the microscope images is 100 μm.

The online version of this article includes the following figure supplement(s) for figure 6:

**Figure supplement 1.** RNA activates RIN14B cells independently of Trpa1.

can activate Piezo1. Instead, fecal/dietary extract sensitivity appears to be linked to other cell line-specific factors found in HEK293 cells but not N2a cells.

## RNA activates RIN14B cells independently of Piezo1

Considering that RNA-sensing by Piezo1 was originally investigated in the gut (*Sugisawa et al., 2020*), we sought to continue our exploration of the effect of ssRNAs in a physiologically relevant cell line, RIN14B. This pancreatic endocrine cell line is commonly used to model gut enterochromaffin cell function and natively expresses Piezo1 (*Nozawa et al., 2009*; *Sugisawa et al., 2020*). To examine the effect of ssRNAs on these cells, we performed calcium imaging in RIN14B cells transfected with GCaMP6s. We measured the change in fluorescence following addition of vehicle, ssRNA40, ssRNA41, fecal RNA, or Yoda1. Each imaging trial was followed by application of ionomycin to determine maximal fluorescence. As expected from previous experiments, the vehicle caused no significant change (*Figure 6A-C*). However, ssRNA40 and ssRNA41 elicited a noticeable calcium response

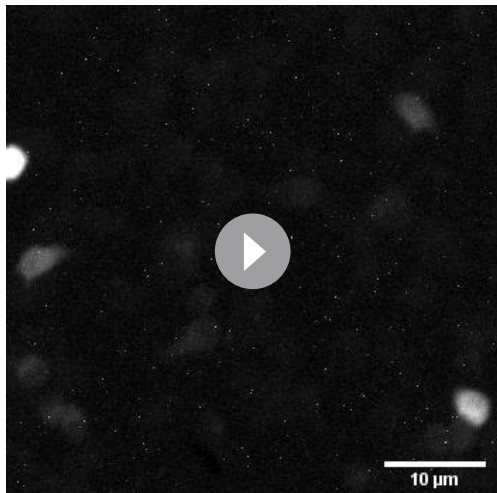

**Video 5.** RIN14B cells respond to fecal RNA. 5 min time lapse of GCaMP6s fluorescence in RIN14B cells during exposure to 25 µg/mL fecal RNA and then 10 µM ionomycin. 1 s of video is equivalent to 30 s of real time.

https://elifesciences.org/articles/83346/figures#video5

in RIN14B cells, unlike in N2a or HEK293 cells (*Figure 6—figure supplement 1*). Fecal RNA also elicited a calcium response, similar in magnitude to that caused by ssRNA40 and ssRNA41 (*Figure 6D-F* and *Video 5*). In comparison, Yoda1 led to a significantly larger increase in fluorescence, consistent with endogenous expression of Piezo1 in RIN14B cells (*Figure 6G-I*).

To investigate the dependency between the RNA-evoked response and Piezo1 function in these cells, we performed calcium imaging on RIN14B cells in the presence of gadolinium. Gadolinium is a broad inhibitor of stretch-activated cation channels including Piezo1 (*Coste et al., 2010*). Cells were exposed first to gadolinium and then to either vehicle, fecal RNA, or Yoda1. At baseline, we noticed general effects of gadolinium on the excitability of RIN14B cells, evident by reduced spontaneous calcium transients during vehicle application (*Figure 6B and C*). Notably, gadolinium did not diminish the response to fecal RNA, but it seemed to have a nonspecific effect on the decay rate of the calcium signal (*Figure 6E and F*). Gadolinium completely abolished the Yoda1 response, confirming successful Piezo1 inhibition (*Figure 6H and I*). Together, these experiments demonstrate that while RIN14B cells do respond to a variety of ssRNAs, their response is not dependent on Piezo1.

In addition to Piezo1, RIN14B cells and gut enterochromaffin cells express the electrophile receptor Trpa1 (*Bellono et al., 2017*; *Nozawa et al., 2009*). We speculated that Trpa1 may be responsible for the RNA-induced calcium response, considering that Trpa1 has been reported to respond to extracellular microRNAs (*Park et al., 2014*). To test this, we performed calcium imaging on RIN14B cells in the presence of a Trpa1 inhibitor, A-967079. As expected, the Trpa1 inhibitor blocked the calcium response following addition of allyl isothiocyanate (AITC), an electrophilic Trpa1 agonist (*Figure 6—figure supplement 1*). However, the Trpa1 inhibitor did not significantly diminish the calcium response following addition of ssRNA40 or ssRNA41 (*Figure 6—figure supplement 1*), indicating that the RNA-induced response is not dependent on Trpa1.

To facilitate a better understanding of the molecular basis for extracellular ssRNA-sensing, we performed single-nuclei RNA sequencing on RIN14B cells. Algorithmic clustering of individual nuclear transcriptomes did not reveal any clear transcriptomic subpopulations. However, there is clearly heterogeneity when examining the prevalence of individual transcripts from cell to cell, which may stem from true biological heterogeneity within the cell line or from technical limitations in our sequencing approach (*Supplementary file 1*). As a starting point for identifying candidate ssRNA receptors, we compiled a gene list based on gene ontology annotations for ion channels, cation transmembrane transporters, and G protein-coupled receptors (*Table 1*). The rough expression prevalence of each gene is conveyed as the fraction of cells in which that gene's transcripts were detected. Additionally, we have made the entire sequencing dataset available as an open resource for investigating RIN14B cell gene expression (Gene Expression Omnibus [GEO], GSE213903). We anticipate these data will be useful for more deeply assessing the fidelity of the RIN14B line as a model of gut enterochromaffin cells.

## Discussion

In this study, we set out to confirm the discovery that Piezo1 is a sensor of fecal microbiome ssRNA. We were not able to detect ssRNA-evoked changes in Piezo1 activity with in vitro calcium imaging and electrophysiological recordings. Instead, we present evidence that ssRNAs and fecal extracts can stimulate calcium influx in cultured cells, but this calcium activity depends on the cell line being used,

**Table 1.** Ion channels and G protein-coupled receptors (GPCRs) in RIN14B cells.

| 'Ion channel activity' genes | Percent of cells expressing gene | 'GPCR activity' genes | Percent of cells expressing gene |
|---|---|---|---|
| Asic1 | 19.5 | Adgra2 | 7.7 |
| Asic2 | 42.7 | Adgra3 | 8 |
| Chrna7 | 7.7 | Adgrb2 | 9.2 |
| Clcn3 | 42.1 | Adgrb3 | 51 |
| Gabrb3 | 16 | Adgrg1 | 16.9 |
| Kcnd3 | 13.5 | Adgrg4 | 15.8 |
| Kcnh2 | 8.6 | Adgrl1 | 22.1 |
| Kcnk3 | 12.3 | Adgrl2 | 37.2 |
| Mcub | 37.5 | Adgrl3 | 61.3 |
| Tmem120a | 16.3 | Adgrv1 | 9.2 |
| Trpa1 | 15.2 | Celsr2 | 9.2 |
|  |  | Celsr3 | 10.9 |
|  |  | Glp1r | 56.4 |

| 'Cation transmembrane transport' genes | Percent of cells expressing gene | | |
|---|---|---|---|
|  |  | Gpr146 | 10.9 |
|  |  | Gpr158 | 53.6 |
|  |  | Gpr176 | 9.7 |
| Ano10 | 10.6 | Gpr6 | 28.7 |
| Atp13a1 | 18.6 | Gprc5b | 14.3 |
| Atp13a3 | 30.7 | Gprc5c | 19.8 |
| Atp1b1 | 17.2 | Grm1 | 23.5 |
| Cnga1 | 8.9 | Lgr4 | 19.8 |
| Grina | 10.6 | Lpar6 | 9.7 |
| Mcoln1 | 16.9 | Oxtr | 8 |
| Nalcn | 9.5 | Tas1r2 | 20.6 |
| Pex5l | 63 | Tm2d1 | 22.9 |
| Piezo1 | 9.2 | Tpra1 | 9.2 |
| Slc29a4 | 9.2 | Vom2r44 | 9.5 |
| Slc30a7 | 38.4 |  |  |
| Slc30a9 | 71.1 |  |  |
| Slc41a2 | 16.6 |  |  |
| Tmem63a | 13.2 |  |  |
| Tmem63b | 35.5 |  |  |
| Tmem63c | 36.4 |  |  |
| Tomm40 | 11.2 |  |  |
| Trpm3 | 49.3 |  |  |
| Trpm7 | 65.9 |  |  |
| Unc80 | 41.3 |  |  |

rather than Piezo1 function. An unexpected finding is that dietary extracts can elicit calcium influx similar to fecal extracts. We also observed differential effects of fecal/dietary extracts between mouse colonies in different facilities, with some preparations showing more or less activity. This highlights the importance of controlling for elements of the diet when working with gut-derived samples.

Our data leave open questions regarding the sources of Piezo1 activation and possible functional roles for ssRNAs in the gut. Interestingly, our observation that the enterochromaffin model cell line RIN14B responds to ssRNAs corroborates the recent evidence that there exists a gut-resident RNA receptor (*Sugisawa et al., 2020*). However, we are unable to reproduce the finding that Piezo1 is an RNA receptor, and we propose that Piezo1 is more likely functioning as a mechanosensor in the gut, as has been shown in many other tissues (*Murthy et al., 2017*; *Syeda, 2021*; *Zhao et al., 2019*).

# Materials and methods

## Key resources table

| Reagent type (species) or resource | Designation | Source or reference | Identifiers | Additional information |
|---|---|---|---|---|
| Strain, strain background (*Mus musculus*) | C57BL/6 | Jax | Strain # 000664 | Maintained in Chesler lab |
| Other (rodent chow) | Dietary extract | LabDiet | Prolab RMH 1800 5LL2 | Autoclaved in Chesler lab and used in Chesler and Patapoutian labs |
| Biological sample (rodent feces) | Fecal extract | This study | Derived from C57BL/6 mice | Freshly isolated in Chesler lab and used in Chesler and Patapoutian labs |
| Cell line (*M. musculus*) | Neuro-2a cells | ATCC | CCL-131 | Maintained in Chesler lab |
| Cell line (*M. musculus*) | Piezo1-KO Neuro-2a cells | Max Delbrück Center for Molecular Medicine; *Moroni et al., 2018* | | Maintained in Chesler lab |
| Cell line (*Homo sapiens*) | HEK293 cells | ATCC | CRL-1573 | Maintained in Chesler lab |
| Cell line (*H. sapiens*) | PIEZO1-KO HEK293 cells | Scripps Research; *Dubin et al., 2017* | | Maintained in Patapoutian lab |
| Cell line (*H. sapiens*) | GCaMP6s HEK293 cells | This study | Derived from the Flp-In T-Rex 293 cell line (Thermo Fisher, R78007) | Maintained in Chesler lab |
| Cell line (*Rattus norvegicus*) | RIN14B cells | ATCC | CRL-2059 | Maintained in Chesler lab |
| Recombinant DNA reagent | CMV-mPiezo1-IRES-eGFP | Addgene | 80925 | Used in Chesler lab |
| Recombinant DNA reagent | CMV-mPiezo1 | This study | Derived from pcDNA5-FRT (Thermo Fisher, V601020) | Used in Chesler lab |
| Recombinant DNA reagent | CMV-hPIEZO1 | This study | Derived in A. Patapoutian lab | Used in Patapoutian lab |
| Recombinant DNA reagent | CMV-GCaMP6s | Addgene | 40753 | Used in Chesler lab |
| Recombinant DNA reagent | pLV-CMV-GCaMP6s-PGK-Hyg | This study | Derived from pLV-CMV-PGK-Hyg (Cellomics Technology, LVR-1046) | Used in Chesler lab |
| Chemical compound, drug | ssRNA40 | Invivogen | A40-41-02 | Used in Chesler and Patapoutian labs |
| Chemical compound, drug | ssRNA41 | Invivogen | A41-41-02 | Used in Chesler lab |
| Chemical compound, drug | Yoda1 | Sigma-Aldrich | SML558-5MG | Used in Chesler and Patapoutian labs |
| Chemical compound, drug | Ionomycin | Sigma-Aldrich | I0634 | Used in Chesler lab |
| Chemical compound, drug | Gadolinium (III) chloride | Sigma-Aldrich | 439770 | Used in Chesler lab |
| Chemical compound, drug | AITC | Sigma-Aldrich | 377430 | Used in Chesler lab |

*Continued on next page*

*Continued*

| Reagent type (species) or resource | Designation | Source or reference | Identifiers | Additional information |
|---|---|---|---|---|
| Chemical compound, drug | A-967979 | Sigma-Aldrich | SML0085 | Used in Chesler lab |
| Chemical compound, drug | Fluo-4 AM dye | Fisher Scientific | F14201 | Used in Chesler lab |
| Chemical compound, drug | Fluo-8 AM dye | AAT Bioquest | 21080 | Used in Patapoutian lab |

## Mice

All experiments involving mice adhered to the animal usage guidelines set by the National Institutes of Health (NIH) and were first approved by the National Institute of Neurological Disorders and Stroke (NINDS) Animal Care and Use Committee. Equal numbers of male/female wildtype C57BL/6J mice between the ages of 6 weeks and 1 year were used. The mice were housed in an AAALAC International accredited pathogen-free facility with ad libitum access to food and water. Water was purified by reverse osmosis and then UV-treated and chlorinated at 15–18 ppm such that after 2 weeks the chlorine concentration was ≥2 ppm. The diet consisted purely of the chemically defined Prolab RMH 1800 (LabDiet, 5LL2) autoclaved rodent chow.

## Cell lines

Cell lines were purchased from the ATCC biobank, acquired by MTA through their origin laboratory, or generated in this study. The following cell lines were used in the study: wildtype N2a cells (ATCC, CCL-131), Piezo1-KO N2a cells (*Moroni et al., 2018*), wildtype HEK293 cells (ATCC, CRL-1573), PIEZO1-KO HEK293 cells (*Dubin et al., 2017*), GCaMP6s HEK293 cells (this study), and RIN14B cells (ATCC, CRL-2059) (*Nozawa et al., 2009*). Cell lines were authenticated by morphological and functional testing (i.e., electrophysiology and calcium imaging). The cells were confirmed to be mycoplasma-free prior to using them in experiments. All cell lines were maintained on polystyrene culture plates (Fisher Scientific, 07-200-80) in a 5% $CO_2$ humidified incubator at 37°C. The growth medium was changed every 2–3 days and consisted of RPMI-1640 (for the RIN14B cells) (Fisher Scientific, 11-875-093) or otherwise DMEM/F12 (Fisher Scientific, 11330032) supplemented with 10% fetal bovine serum (Fisher Scientific, 26140079). Cells were passaged when they reached confluence, which was roughly twice per week, and all cultures that were used for experiments were not propagated beyond 20 passages. For passaging, cells were rinsed in PBS (Fisher Scientific, 10010023) and then incubated in Accutase (Fisher Scientific, 00-4555-56) for ~5 min at 37°C to detach. Cells were collected in a 15 mL tube (Fisher Scientific, 12-565-268) and centrifuged at 300 rcf for 3 min to pellet. The supernatant was aspirated, and cells were resuspended in growth medium followed by plating in new polystyrene plates. Typical dilution ratios for passaging were between 1:3 and 1:20.

## Generating GCaMP6s stable HEK293 cells

GCaMP6s was generated by a custom gene synthesis service (Epoch Life Science) and subcloned into a pLV-CMV-PGK-Hyg lentiviral vector (Cellomics Technology, LVR-1046) to make pLV-CMV-GCaMP6s-PGK-Hyg. This vector was used to produce lentiviral particles (Vigene Biosciences). Then, wildtype HEK293 cells (ATCC, CRL-1573) were infected in regular growth medium with 5 µg/mL polybrene (Sigma-Aldrich, TR-1003-G) and the lentivirus at a multiplicity of infection of five viral particles per cell. Transduction was allowed to occur overnight. The cells were then replated, and 48 hr later 100 µg/mL Hygromycin B was added to initiate antibiotic selection. After 2 weeks of culture and passaging, a hygromycin-resistant polyclonal GCaMP6s stable HEK293 cell line was isolated.

## Plasmid transfection

Wildtype HEK293, GCaMP6s HEK293, Piezo1-KO N2a, and RIN14B cells were used in transfection experiments. Other cell lines were used untransfected to examine endogenously expressed Piezo1 (wildtype N2a cells) or were used in FLIPR experiments described further below (PIEZO1-KO HEK293 cells). Cells were seeded in 24-well plates 24–72 hr before transfection. Transfection was performed when the cells were at ~70% confluence using 500 ng plasmid DNA and the Lipofectamine 3000 kit

(Fisher Scientific, L3000001) following the manufacturer's instructions. The following plasmids were used in the study: CMV-mPiezo1-IRES-eGFP, CMV-mPiezo1, and CMV-GCaMP6s. CMV-mPiezo1-IRES-eGFP was a gift from Ardem Patapoutian (Addgene plasmid #80925; http://n2t.net/addgene:80925) and was used for all electrophysiological recordings. CMV-mPiezo1 was previously generated in-house by subcloning mouse Piezo1 into the pcDNA5/FRT expression vector (Fisher Scientific, V601020) – this plasmid was used for all calcium imaging of Piezo1 activity. CMV-GCaMP6s was a gift from Douglas Kim and the GENIE Project (Addgene plasmid #40753; http://n2t.net/addgene:40753) and was used for calcium imaging of RIN14B cells.

## Calcium imaging

Ringer's solution was used for all physiological assays, consisting of 133 mM NaCl, 3 mM KCl, 2.5 mM $CaCl_2$, 1 mM $MgCl_2$, 10 mM glucose, 10 mM HEPES, and 40.9 mM sucrose (all from Sigma-Aldrich) dissolved in water. The pH was adjusted to 7.3 with 1 M NaOH, and the osmolality was ~330 mmol/kg. Calcium influx was visualized in N2a cells and HEK293 cells using Fluo-4 AM dye (Fisher Scientific, F14201) or the GCaMP6s HEK293 cell line described above. For Fluo-4 AM imaging, 50 µg Fluo-4 AM was dissolved in 44 µL DMSO (Sigma-Aldrich, D2650) and mixed with 9 µL Pluronic F-127 (Fisher Scientific, P-3000MP) by vortexing. Then, 50 µL of this mixture was diluted in 14.3 mL Ringer's solution to make the 'loading solution.' Cells cultured in eight-chamber slides (Fisher Scientific, 177445PK) were first rinsed with Ringer's solution and then incubated in loading solution for 1 hr light-protected at room temperature. After 1 hr, the loading solution was removed, the cells were rinsed with Ringer's solution, and then immediately imaged in Ringer's solution using a pco.panda sCMOS back-illuminated camera at 3 frames/s with an Olympus IX73 inverted microscope and ×10 air objective. Videos were recorded and saved using pco.camware software. Solutions containing different compounds were added and removed via micropipette during video recording while maintaining the same volume (150 µL) in the chamber. Stock solutions of all compounds were dissolved and prepared following the manufacturer's instructions, and the final concentration used in each experiment can be found in each figure legend. The following commercially available compounds were used in the study: ssRNA40 (Invivogen, A40-41-02), ssRNA41 (Invivogen, A41-41-02), Yoda1 (Sigma-Aldrich, SML558-5MG), AITC (Sigma-Aldrich, 377430), gadolinium(III) chloride (Sigma-Aldrich, 439770), A-967979 (Sigma-Aldrich, SML0085), and ionomycin (Sigma-Aldrich, I0634).

Fiji software was used to import and analyze video files from pco.camware software (pco). The Template Matching and Slice Alignment plugin was used to align all video frames to correct for any drift (*Tseng et al., 2011*). For creating still-frame images, a grayscale baseline image was generated from an average of the first 10 frames of the recording. Separately, a fire-scale standard deviation Z-projection was generated from the frames where the cells were exposed to a specific treatment. The Z-projection image was then overlaid on top of the grayscale baseline image to visualize which cells responded to a given treatment. Supplemental videos were made from the raw recordings and exported at 10 frames/s so that 1 s of video is equivalent to 30 s of real time.

For quantifying responses, cellular regions of interest (ROIs) were drawn around randomly selected individual cells and used to measure the mean pixel intensity per frame. In cases where only certain regions within the field of view showed calcium influx, such as with poor fluid dispersion or the variable activation seen with fecal/dietary extracts, the ROI selection was restricted to this region of activation. We found that dissolved ssRNA40 and crude extracts produced substantial autofluorescence, which could be a confounding factor when analyzing calcium imaging recordings. However, a standard image background subtraction procedure effectively eliminated most of the fluorescence artifacts. This was accomplished by drawing an additional set of 10 background ROIs per recording that were in cell-free areas in the field of view. The mean pixel intensities were exported to Microsoft Excel software for normalization and quantification. An average of the background ROI values was subtracted from each cellular ROI frame-by-frame to correct for artifactual changes in background fluorescence. These values were then used to calculate the $\Delta F/F_0$ for each cell. The mean pixel intensity of the first 10 frames was averaged to yield $F_0$, and then $F_0$ was subtracted from each frame's pixel intensity on a frame-by-frame basis to determine $\Delta F$. Dividing $\Delta F$ by $F_0$ ($\Delta F/F_0$) normalized each cell's change in calcium fluorescence to its baseline level of fluorescence.

To quantify the maximal response to different treatments, the peak $\Delta F/F_0$ from within the treatment exposure timeframe was selected for each cell. All recordings ended with ionomycin treatment to elicit

maximum calcium influx for normalization purposes. In cases where two different recordings showed significantly different ionomycin responses, the $\Delta F/F_0$ for experimental treatments was normalized a percentage of the peak ionomycin response. $\Delta F/F_0$ values were exported to GraphPad 8.0 (Prism) for visualization and graphing. An n = 25–50 cells was analyzed for each calcium imaging recording and are representative of at least three independently performed transfections and recording sessions per condition.

## Electrophysiology

N2a cells and HEK293 cells were plated in 35 mm dishes (Fisher Scientific, 353001) ~24 hr prior to recording, and the next day they were rinsed once in Ringer's solution before recording in Ringer's solution. Patch-clamp recordings were performed in whole-cell voltage-clamp mode by glass micro-pipette electrodes that were pulled and polished to 2–6 MΩ resistance. The pipette was filled with internal solution consisting of 133 mM CsCl, 1 mM $CaCl_2$, 1 mM $MgCl_2$, 5 mM EGTA, 10 mM HEPES, 4 mM Mg-ATP, 0.4 mM $Na_2$-GTP, 43.8 mM sucrose (all from Sigma-Aldrich). Internal solution pH was adjusted to 7.3 with 1 M CsOH, and the osmolality was ~320 mmol/kg. After establishing a GΩ seal with the patch pipette on a cell membrane and breaking into whole-cell configuration, cells were held at –80 mV and mechanically stimulated with a separate glass-polished probe to elicit Piezo1 currents. The probe was a micropipette that was heat-polished to seal the tip until rounded with a width of 3–5 µm. The probe was attached to a piezoelectric translator (Physik Instrumente, P841.20) and mounted on a micromanipulator (Sutter Instrument, MP-225) at 45° angle to the cell surface. To stimulate the cells, the probe was maneuvered to rest ~1 µm above the cell surface and then sequentially indented for 200 ms in 1 µm increments from 1 to 10 µm with a 2 ms ramp time. Each indentation was separated by 2 s. Whole-cell currents were measured by a Multiclamp 700b amplifier (Molecular Devices) and digitized by a Digidata 1550 (Molecular Devices) at 100 kHz and then low-pass filtered at 10 kHz. The signals were saved digitally using Clampex 11.1 software (Molecular Devices).

Clampfit 11.1 software (Molecular Devices) was used to analyze the electrophysiological recordings. Any whole-cell recording showing a static leak current >200 pA was discarded from analysis due to poor patch seal quality. Additionally, cells with a peak mechanically evoked current <30 pA were considered nonresponders and discarded since these currents are near the baseline noise level and their kinetics could not be reliably analyzed. Additionally, cells with a peak current >4000 pA were discarded due to the abnormally high values and generally unhealthy swelled morphology of such cells. Finally, recordings were discarded if the patch pipette seal broke before three consecutive mechanically evoked responses because low indentation responses have distinct kinetics that bias analysis. In the end, an equal number of recordings (5–7) were discarded from each condition (from a total of 15–25 attempted cells/condition), with no apparent systematic bias toward any of the control or ssRNA conditions. The remaining recordings (one per cell) were filtered at 1 kHz and thresholded to 0 pA. The maximal current was measured by the largest amplitude response before patch breakage or by reaching 10 µm membrane indentation, whichever came first. This same response was then used to approximate the time constant of inactivation (tau) by calculating the time taken to decay 63.2% back to baseline. The mechanical activation threshold was determined by the level of membrane indentation (µm) to elicit the first current response peak (pA) above the baseline level of noise. No systematic differences were observed for baseline noise level or maximum membrane indentation between conditions. Values were exported from Clampfit 11.1 to GraphPad 8.0 (Prism) for visualization and graphing. A minimum of n = 7 cells were analyzed per recording condition.

## FLIPR assay

PIEZO1-KO HEK293 or wildtype HEK293 cells were grown in Dulbecco's modified Eagle's medium containing 4.5 mg/mL glucose, 10% fetal bovine serum, and 1× pen/strep. Cells were plated in 6-well plates and transfected using Lipofectamine 2000 (Thermo Fisher Scientific) according to the manufacturer's instructions. Human PIEZO1 fused to IRES-TdTomato or mouse Piezo1 fused to IRES-GFP was transfected at 2 µg per well (6-well plate) for FLIPR. One day after transfection, the cells were dissociated from 6-well plates with trypsin and re-seeded into a 384-well plate, at 20,000 cells per well. The plate was then cultured for 1 day before washing with assay buffer (1× HBSS, 10 mM HEPES, pH 7.4) in a ELx405 CW plate washer (BioTek Instruments). The cells were then incubated with 1.25 µM calcium indicator Fluo-8 AM (AAT Bioquest) in the assay buffer at 37°C for 1 hr. After washing out

excess dye, fluorescence was measured on a FLIPR Tetra upon treatment with various reagents. A 1 mM stock solution of Yoda1 in dimethyl sulfoxide (DMSO) was used, resulting in a final concentration of 5 µM Yoda1 and 0.5% DMSO in the assay. The effect of ssRNA40 was tested at concentrations of 5, 2.5, 1.25, and 0.625 µg/mL. Ionomycin was added to 10 µM concentration as a final normalization. All measurements were taken from four biological replicates (four different wells in 384-well plate).

## Crude extract preparations

Fresh mouse feces were gathered by gently holding the mouse over a sterile 1.5 mL tube and collecting the fecal matter directly into the tube as it was excreted. Feces from 10 to 20 adult mice were pooled together, diluted to 0.1 g/mL in Ringer's solution, and homogenized using a sterile mortar and pestle. The sample was then centrifuged at 300 rcf for 3 min to pellet any remaining undissolved fecal matter and then sequentially filtered through 100, 40, and finally 0.45 µm mesh membranes (Fisher Scientific, SLHAR33SS) to produce the 'fecal extract.' Because this extract was strongly autofluorescent during calcium imaging, the fecal extract was further diluted 1:20 in Ringer's solution from its original 0.1 g/mL to a final concentration of 5 mg/mL when applying it to cell cultures. Dietary extracts were prepared in identical fashion to fecal extracts, with the exception that mouse food pellets were first crushed in a dry state using mortar and pestle and then transferred to Ringer's solution (0.1 g/mL).

Four different fecal extract preparations and three different dietary extract preparations were independently made and tested over the course of the study at the NIH. A separate set of fecal and dietary extracts were prepared from mice at Scripps/HHMI. The NIH-sourced extracts showed substantially more stimulatory activity on HEK293 cells when tested in parallel with the Scripps/HHMI extracts, indicating possible differences owing to the specific mouse colony and commercial diet source. For RNase treatment of fecal extracts, RNase A (Fisher Scientific, EN0531) was added at a final concentration of 500 µg/mL to the 0.1 g/mL fecal extract and incubated at 37°C for 30 min. Mock-treated fecal extracts were handled in the same way but without addition of RNase A. The RNase- and mock-treated fecal extracts were then used for calcium imaging on Piezo1-transfected HEK293 cells at a final concentration of 5 mg feces per mL.

To check whether the fecal and dietary extracts were nonspecifically activating cells due to changes in osmolality or pH, these properties were examined in extracts that were diluted to the working concentration of 5 mg/mL in Ringer's solution. The control Ringer's solution that was tested had an osmolality of ~336 mmol/kg and 7.5 pH. In comparison, fecal extract was ~332 mmol/kg and 7.5 pH, and dietary extract was ~334 mmol/kg and 7.5 pH. These measurements indicate that the extracts did not substantially affect the osmolality or pH of the solutions.

## Fecal RNA purification

RNA was extracted from mouse feces by a standard phenol/chloroform protocol. Then, 500 µL TRIzol (Fisher Scientific, 15596026) was added per 50 mg feces and homogenized using an RNase-free tube and plunger (Takara, 9791A). Also, 100 µL chloroform (Sigma-Aldrich) was then added per 500 µL TRIzol, vortexed vigorously followed by 3 min incubation at room temperature, and centrifuged at 12,000 rcf for 10 min at 4°C. The aqueous layer was transferred to a clean tube and the RNA was extracted using the miRNeasy kit (QIAGEN, 217004) following the manufacturer's instructions. The RNA content of the samples was measured using a NanoDrop (Fisher Scientific, ND-2000), confirming an ~2.0 ratio of 260/280 nm absorbance. These purified RNA samples yielded ~300 ng/µL RNA, while RNA in the crude fecal extracts was below the detection range of the NanoDrop. The samples were separated on an agarose gel and examined for nucleic acid content. The purified fecal RNA manifested as a smear, ranging from short oligonucleotides tens of base pairs (bp) in size up to 300 bp (*Figure 3C*). To confirm that the samples were in fact RNA and not DNA, fecal extracts and purified fecal RNA were treated with 500 µg/mL RNase A (Fisher Scientific, EN0531) for 30 min at 37°C before running on an agarose gel (*Figure 3C*). The fecal RNA was then tested on Piezo1-transfected HEK293 cells at a final concentration of 10 µg/mL, which was the same concentration used in the Sugisawa study (*Figure 3D*).

Additionally, fecal RNA was purified using the same methodology and NucleoSpin TriPrep kit (Macherey-Nagel, 740966.10) as in the Sugisawa study, following the manufacturer's instructions. Then, 350 µL buffer RP1 was added per 50 mg feces, and the samples were homogenized using an RNase-free tube and plunger (Takara, 9791A) and vortexed for 5 s. A wide-bore pipette was then used

to transfer the samples to a NucleoSpin filter, and the RNA was washed and extracted following the TriPrep protocol. RNA concentration was 100–200 ng/μL, and the purity was confirmed by an ~2.0 ratio of 260/280 nm absorbance via NanoDrop. Applying these purified fecal RNA samples at a final concentration of 10 μg/mL did not elicit calcium influx in Piezo1-transfected HEK293 cells.

## Single-nuclei RNA sequencing

RIN14B cells were put on ice, washed with chilled PBS, and then lysed with chilled Nuclei EZ lysis buffer (Sigma-Aldrich, NUC-101). Single cells were isolated with a 40 μm filter and pelleted in a centrifuge for 8 min, 800 rcf, 4°C. The nuclei were resuspended using PBS with 1% BSA and counted using a hemocytometer with trypan blue viability dye. The nuclei were centrifuged and resuspended at an appropriate volume for the 10X Chromium system (10X Genomics). The nuclei were counted once more to check the number and quality before proceeding with 10X Chromium processing and library construction as per the manufacturer's instructions. Next Gen sequencing with a Chromium V2 chemistry was carried out on an Illumina NextSeq 500. Illumina NextSeq 500 pre-mRNA sequencing data were aligned to the *Rattus norvegicus* genome using CellRanger. The data were then analyzed with Seurat V3.0 as described previously (*Butler et al., 2018*).

## Statistical analysis

All data were first tested for normality using the Shapiro–Wilk test. Normally distributed data were analyzed by one-way ANOVA with Bonferroni multiple-comparisons correction, and non-normally distributed data were analyzed by Kruskal–Wallis test with Dunn's multiple-comparisons correction. Statistical significance was determined by a p-value<0.05. The degree of statistical significance is indicated in each figure legend using asterisks. One cell equals one biological replicate for calcium imaging and electrophysiology experiments. The *n* number of biological replicates for each condition are representative of at least three separately run experiments. The *n* and error bar definitions are reported in each figure legend. No power analyses were done to determine sample sizes a priori, but our sample sizes adhere to those reported in similar previous studies (*Sugisawa et al., 2020*). All graphing and statistical testing was performed using GraphPad 8.0 software (Prism), and figures were assembled using Adobe Illustrator 2021.

# Acknowledgements

We thank Gary Lewin for generously providing the Piezo1-KO N2a cell line and the Janelia GENIE Project for the GCaMP6s sequence. Nick Ryba and Minh Nguyen performed the sample preparation and analysis of the single-nuclei RNA sequencing of RIN14B cells. We are grateful to Shang Ma, Chuan Wu, Jialie Luo, and Aisha AlJanahi for technical assistance in the study. The research was funded by the National Center for Complementary and Integrative Health (NCCIH) and National Institute of Neurological Disorders and Stroke (NINDS) Intramural Research Programs (ATC); the Howard Hughes Medical Institute and NIH R35 NS105067 (AP); and the National Center for Advancing Translational Sciences (NCATS) through the National Institutes of Health (NIH) Helping to End Addiction Long-term[SM] (HEAL) Initiative. The content is solely the responsibility of the authors and does not necessarily represent the official views of the NIH or its HEAL initiative.

# Additional information

## Competing interests

Alexander T Chesler: Reviewing editor, eLife. The other authors declare that no competing interests exist.

## Funding

| Funder | Grant reference number | Author |
|---|---|---|
| National Center for Complementary and Integrative Health | Intramural funds | Alexander T Chesler |
| National Institute of Neurological Disorders and Stroke | Intramural funds | Alexander T Chesler |
| National Center for Advancing Translational Sciences | Intramural funds | Alexander T Chesler |
| Howard Hughes Medical Institute | | Ardem Patapoutian |
| National Institutes of Health | R35 NS105067 | Ardem Patapoutian |

The funders had no role in study design, data collection and interpretation, or the decision to submit the work for publication.

## Author contributions

Alec R Nickolls, Conceptualization, Data curation, Formal analysis, Validation, Investigation, Visualization, Methodology, Writing – original draft, Project administration, Writing – review and editing; Gabrielle S O'Brien, Data curation, Formal analysis, Investigation, Methodology, Writing – original draft, Writing – review and editing; Sarah Shnayder, Yunxiao Zhang, Data curation, Formal analysis, Investigation, Methodology, Writing – review and editing; Maximilian Nagel, Resources, Investigation, Methodology, Writing – review and editing; Ardem Patapoutian, Conceptualization, Resources, Data curation, Supervision, Investigation, Methodology, Project administration, Writing – review and editing; Alexander T Chesler, Conceptualization, Resources, Data curation, Supervision, Funding acquisition, Investigation, Methodology, Writing – original draft, Project administration, Writing – review and editing

## Author ORCIDs

Alec R Nickolls http://orcid.org/0000-0002-7399-4304
Ardem Patapoutian http://orcid.org/0000-0003-0726-7034
Alexander T Chesler http://orcid.org/0000-0002-3131-0728

## Ethics

This study was performed in strict accordance with the recommendations in the Guide for the Care and Use of Laboratory Animals of the National Institutes of Health. All of the animals were handled according to approved institutional animal care and use committee (IACUC) protocols (#1365) of the NINDS-IRP.

## Decision letter and Author response

Decision letter https://doi.org/10.7554/eLife.83346.sa1
Author response https://doi.org/10.7554/eLife.83346.sa2

# Additional files

## Supplementary files

• Transparent reporting form

• Supplementary file 1. RIN14B single-nuclei RNA sequencing results. A list of all gene transcripts detected in RIN14B cells and their prevalence in the cell population with a weak cutoff limit.

## Data availability

All data generated or analyzed during this study are included in the manuscript and supporting files. Sequencing data have been deposited on the GEO website.

The following dataset was generated:

| Author(s) | Year | Dataset title | Dataset URL | Database and Identifier |
|---|---|---|---|---|
| Nickolls AR, Chesler AT | 2022 | Reevaluation of Piezo1 as a gut RNA sensor | https://www.ncbi.nlm.nih.gov/geo/query/acc.cgi?acc=GSE213903 | NCBI Gene Expression Omnibus, GSE213903 |

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
