## [Editor Report]

This is an important study that resolves a controversy about a proposed molecular linkage between the fields of mechanobiology and RNA signaling. While prior research had claimed that a specific mechanosensitive ion channel in the gut responds to a specific fecal RNA, this study provides compelling evidence that the mechanosensitive ion channel does not respond to the RNA.

---

## [Decision Letter]

**Decision letter after peer review:**

Thank you for submitting your article "Reevaluation of Piezo1 as a gut RNA sensor" for consideration by *eLife*. Your article has been reviewed by 3 peer reviewers, and the evaluation has been overseen by a Reviewing Editor and Richard Aldrich as the Senior Editor. The reviewers have opted to remain anonymous.

The reviewers have discussed their reviews with one another, and the Reviewing Editor has drafted this to help you prepare a revised submission. Please consider the suggestions made by reviewers #1 and #3.

*Reviewer #1 (Recommendations for the authors):*

Ideally, this article should be published in Cell (in response to Sugisawa et al., 2020), but *eLife* is also a very appropriate place for it. I have only a few suggestions to improve what I find is already a very good manuscript.

1. Figures 1 and 4 should include panels displaying the peak current vs the poke indentation displacement to better illustrate the various stimulus-response relationships.

2. Figure 1b should include a raw trace for the vehicle treatment condition, in addition to the raw traces for the ssRNA40 and ssRNA41 treatments.

3. It would be a much better resource for future work aimed at identifying the rector for ssRNA, if the authors obtain transcriptomic profiles for ssRNA-sensing RIN14b cells and the non-ssRNA sensitive N2A cells, and compute genes that are differentially expressed. In addition, it would help to not limit this candidate list to known receptors, but to extend it to all putative transmembrane proteins (e.g. 2 predicted TM domains). This request is not within the major scope of this work, but please consider doing it.

4. In the final paragraph of the results the authors state that "algorithmic clustering of individual nuclear transcriptomes revealed a homogenous cell culture population without any meaningful transcriptomic sub-populations", but then analyze expression levels "as a fraction of cells". This seems contradictory and confusing. Please clarify whether or not the population of cells is indeed homogenous and/or why this analysis needs to be done 'as a fraction of cells'.

*Reviewer #2 (Recommendations for the authors):*

Using calcium imaging and electrophysiology, the authors in this study have systematically examined the previously reported finding that the mechanical activation ion channel Piezo1 might be directly activated by a gut RNA sensor. The experiments have been carefully designed and proper controls have been included. It is compelling that the present study disputed the previous finding that ssRNA can specifically activate Piezo1 independent of mechanical stimulation. The conclusion from the present study is well supported by the data. The reviewer supports the publication of the current manuscript.

*Reviewer #3 (Recommendations for the authors):*

I have only two concerns:

In the abstract and introduction, the authors refer to Sugisawa et al's finding regarding ssRNA4 and Piezo1 interaction as "provocative". I agree that the report was met with great skepticism, but was it really provocative? It is entirely up to the authors to keep the wording as is though.

Line 66 – "it was shown that Piezo1 in the mouse gut is not activated by mechanical forces" to my recollection, the authors of the disputed study never claimed that Piezo1 is not activated by force, they maintained, erroneously, that Piezo1 is activated by ssRNA. Please clarify.

---

## [Author Response]

The reviewers have discussed their reviews with one another, and the Reviewing Editor has drafted this to help you prepare a revised submission. Please consider the suggestions made by reviewers #1 and #3.Reviewer #1 (Recommendations for the authors):Ideally, this article should be published in Cell (in response to Sugisawa et al., 2020), but eLife is also a very appropriate place for it. I have only a few suggestions to improve what I find is already a very good manuscript.1. Figures 1 and 4 should include panels displaying the peak current vs the poke indentation displacement to better illustrate the various stimulus-response relationships.

We have updated figures 1 and 4, legends, and result text to include these suggestions.

2. Figure 1b should include a raw trace for the vehicle treatment condition, in addition to the raw traces for the ssRNA40 and ssRNA41 treatments.

We have updated figure 1, legend, and result text to include this suggestion.

3. It would be a much better resource for future work aimed at identifying the rector for ssRNA, if the authors obtain transcriptomic profiles for ssRNA-sensing RIN14b cells and the non-ssRNA sensitive N2A cells, and compute genes that are differentially expressed. In addition, it would help to not limit this candidate list to known receptors, but to extend it to all putative transmembrane proteins (e.g. 2 predicted TM domains). This request is not within the major scope of this work, but please consider doing it.

Because our work was focused on clarifying Piezo1’s putative role as an RNA receptor, we have not performed additional sequencing experiments to refine the list of candidate RNA receptor genes. However, we agree this is an important future venture and believe our inclusion of the entire RIN14B transcriptome list (Table S1) is a good step toward this goal.

4. In the final paragraph of the results the authors state that "algorithmic clustering of individual nuclear transcriptomes revealed a homogenous cell culture population without any meaningful transcriptomic sub-populations", but then analyze expression levels "as a fraction of cells". This seems contradictory and confusing. Please clarify whether or not the population of cells is indeed homogenous and/or why this analysis needs to be done 'as a fraction of cells'.

We agree there is clearly heterogeneity in transcriptome expression from cell to cell in our dataset that may come from true biological differences as well as technical limitations to the nuclear RNA sequencing sensitivity. We updated the language in the Results to reflect these caveats.

Reviewer #3 (Recommendations for the authors):I have only two concerns:In the abstract and introduction, the authors refer to Sugisawa et al's finding regarding ssRNA4 and Piezo1 interaction as "provocative". I agree that the report was met with great skepticism, but was it really provocative? It is entirely up to the authors to keep the wording as is though.

We have removed all uses of the term “provocative” in the abstract and introduction.

Line 66 – "it was shown that Piezo1 in the mouse gut is not activated by mechanical forces" to my recollection, the authors of the disputed study never claimed that Piezo1 is not activated by force, they maintained, erroneously, that Piezo1 is activated by ssRNA. Please clarify.

We have removed the sentence in question since it is confusing and does not add useful information to the introduction.